# Reactive Astrocytosis—A Potential Contributor to Increased Suicide in Long COVID-19 Patients?

**DOI:** 10.3390/brainsci14100973

**Published:** 2024-09-27

**Authors:** Alessandra Costanza, Andrea Amerio, Andrea Aguglia, Martina Rossi, Alberto Parise, Luca Magnani, Gianluca Serafini, Mario Amore, Daniel Martins, Khoa D. Nguyen

**Affiliations:** 1Department of Psychiatry, Faculty of Medicine, University of Geneva (UNIGE), 24 Rue du Général-Dufour, 1211 Geneva, Switzerland; 2Department of Psychiatry, Faculty of Biomedical Sciences, University of Italian Switzerland (USI), Via Giuseppe Buffi 13, 6900 Lugano, Switzerland; 3Department of Psychiatry, Adult Psychiatry Service, University Hospitals of Geneva (HUG), Rue Gabrielle-Perret-Gentil 4, 1205 Geneva, Switzerland; 4“Nel Chiostro”, Medical and Study Center, Via Camillo Leone 29, 13100 Vercelli, Italy; 5Section of Psychiatry, Department of Neuroscience, Rehabilitation, Ophthalmology, Genetics, Maternal and Child Health, University of Genoa, Via Balbi, 5, 16132 Genoa, Italy; andrea.amerio@unige.it (A.A.); andrea.aguglia@unige.it (A.A.); gianluca.serafini@unige.it (G.S.); mario.amore@unige.it (M.A.); 6IRCCS Polyclinic Hospital San Martino, Largo Rosanna Benzi 10, 16132 Genoa, Italy; 7Geriatric-Rehabilitation Department, University Hospital of Parma, 43126 Parma, Italy; aparise@ao.pr.it; 8Department of Psychiatry, San Maurizio Hospital of Bolzano, Via Lorenz Böhler, 5, 39100 Bolzano, Italy; magnani1991@gmail.com; 9Department of Neuroimaging, Institute of Psychiatry, Psychology and Neuroscience (IoPPN)—King’s College London (KCL), Strand Campus, London WC2R 2LS, UK; daniel.martins@kcl.ac.uk; 10NIHR Maudesley BRC, 16 De Crespigny Park, SE5 8AF South London and Maudesley NHS Trust, Denmark Hill, London SE5 8AZ, UK; 11Program in Immunology, Stanford University, 450 Jane Stanford Way, Stanford, CA 94305, USA; khoa.d.nguyen@gmail.com; 12Department of Biomedical Sciences, Chinese University of Hong Kong, Hong Kong SAR, China

**Keywords:** long COVID-19, depression, suicidal ideation, suicidal behavior, astrocytes, reactive astrocytosis, blood–brain barrier, neuroinflammation

## Abstract

Background: Long COVID-19 is an emerging chronic illness of significant public health concern due to a myriad of neuropsychiatric sequelae, including increased suicidal ideation (SI) and behavior (SB). Methods: This review provides a concise synthesis of clinical evidence that points toward the dysfunction of astrocytes, the most abundant glial cell type in the central nervous system, as a potential shared pathology between SI/SB and COVID-19. Results: Depression, a suicide risk factor, and SI/SB were both associated with reduced frequencies of various astrocyte subsets and complex proteomic/transcriptional changes of astrocyte-related markers in a brain-region-specific manner. Astrocyte-related circulating markers were increased in depressed subjects and, to a less consistent extent, in COVID-19 patients. Furthermore, reactive astrocytosis was observed in subjects with SI/SB and those with COVID-19. Conclusions: Astrocyte dysfunctions occurred in depression, SI/SB, and COVID-19. Reactive-astrocyte-mediated loss of the blood–brain barrier (BBB) integrity and subsequent neuroinflammation—a factor previously linked to SI/SB development—might contribute to increased suicide in individuals with long COVID-19. As such, the formulation of new therapeutic strategies to restore astrocyte homeostasis, enhance BBB integrity, and mitigate neuroinflammation may reduce SI/SB-associated neuropsychiatric manifestations among long COVID-19 patients.

## 1. Introduction

Since its emergence as a cluster of lethal respiratory illnesses in China at the end of 2019, over 770 million people globally have had the coronavirus disease 2019 (COVID-19) (as of 21 January 2024) [1]. The acute impact of COVID-19, characterized by respiratory symptoms and inflammation-induced multi-organ dysfunction, has been extensively studied and appears to be effectively controlled by various therapeutic approaches. However, recent clinical studies highlighted an increase in COVID-19-associated co-morbidities after the resolution of acute disease symptoms [2]. Notably, long COVID-19 patients often experience neurological anomalies, including fatigue, anxiety, and cognitive impairment [3]. These complications have ignited scientific and clinical interest in understanding the long-term neurotropic impact of SARS-CoV-2 infection.

Suicidal ideation (SI) and suicidal behavior (SB), including suicidal attempts (SA), have been considered as neurological sequelae of long COVID-19 [4]. However, existing studies of COVID-19 patients during the acute disease phase have not yet reached a consensus on the potential increase in suicide prevalence due to SARS-CoV-2 infection. Several studies (*n* < 1000) conducted in the US, China, and Slovenia, involving different demographic groups, such as veterans, healthcare workers, and young adults, have established a connection between acute COVID-19 and an elevated SI prevalence [5,6,7]. However, a more extensive study conducted in Denmark (*n* > 250,000) failed to confirm this association between SB and COVID-19 [8].

Despite these discrepancies, recent studies suggested that an increase in SI/SB may manifest as a neuropsychiatric phenomenon during the post-acute phase of COVID-19 [9]. Given a possible overlap between hyperinflammation in COVID-19 and SI/SB development [10], factors contributing to the propagation of chronic neuroinflammation might be associated with increased suicide prevalence among long COVID-19 patients. In this regard, various effector immune cell populations, such as monocytes, macrophages, T cells, and microglia, have been implicated in the orchestration of inflammatory responses in COVID-19. Emerging evidence also suggested that astrocytes, a glial cell type, may play a role in the orchestration of COVID-19-associated neuroinflammation [11].

Therefore, the purpose of this narrative review is to highlight significant clinical evidence of astrocyte dysfunction as a potential shared pathology in the development of SI/SB and COVID-19. Additionally, we propose that disruption in the astrocyte regulation of the blood–brain barrier (BBB) integrity may contribute to chronic neuroinflammation and, consequently, the development of depression and SI/SB in individuals experiencing long-term effects of SARS-CoV-2 infection.

## 2. Astrocytes and CNS Homeostasis

Astrocytes, the most abundant cell type in the CNS, differ from microglia in their developmental origin. While microglia arise from myeloid progenitors, astrocytes develop from neural precursor cells in various neurogenic hot spots in the CNS [12]. The regulation of astrogenesis primarily involves JAK-STAT signaling, which determines the molecular characteristics of glial cells by controlling the expression of Glial Fibrillary Acidic Protein (GFAP), S100 Calcium-Binding Protein B (S100B), Sex-determining Region Y-box 9 (SOX9), Aldehyde Dehydrogenase 1 Family Member A1 (ALDH1), Glutamate Aspartate Transporter (GLAST), and other markers [13]. Of note, GFAP is specifically expressed by astrocytes in the CNS and exhibits intense staining in response to pathological insults [14]. Various subtypes of astrocytes have been identified, including gray matter protoplasmic/fibrous astrocytes, cerebellar and olfactory bulb velate astrocytes, radial astrocytes, perivascular/marginal astrocytes, and ependymocytes [15]. While early studies suggested that astrocytes were accessory cells, merely providing passive support to electrically active neurons, accumulating evidence now indicates their vital and active role in various physiological functions of the CNS. Structurally, astrocytes actively participate in shaping and enhancing neural networks and contribute to the formation and maintenance of the BBB [16]. Metabolically, astrocytes serve a dual function by supplying energy to neurons through the “lactate shuttle” and employ the glutamine–glutamate cycle to prevent glutamate-induced neurotoxicity [17,18]. Furthermore, astrocytes secrete a plethora of neurotrophic factors and participate in CNS waste clearance through the glymphatic system [19,20].

In addition to their role in maintaining CNS homeostasis, astrocytes respond to infectious stimuli or injuries within the CNS by transitioning into a reactive state. While reactive astrocytosis resolves with the recession of protective neuroinflammation, chronic CNS inflammatory response could trigger the transduction of neuroinflammatory signals from reactive astrocytes to pericytes [21]. This sustained inflammatory network ultimately results in endothelial cell dysfunction and loss of BBB integrity [22]. Consequently, chronic reactive astrocytosis has been implicated in a wide range of neurobiological and neuropsychiatric illnesses [23].

## 3. Astrocyte Dysfunction in Suicide Development

Clinical evidence for various astrocyte perturbations in suicide development is derived primarily from studies involving suicidal subjects and individuals with depression, a major risk factor for suicide. While the details of such studies have been extensively reviewed elsewhere [24,25], this section aims to provide a concise distillation of relevant findings into a mechanistic framework that enables further hypothesis-driven clinical exploration of the role of astrocytes in suicide development.

### 3.1. Astrocyte-Related Abnormality in Depressed Subjects

The distribution of astrocytes has been examined in post-mortem brain tissues of patients experiencing major depressive disorder (MDD), a major risk factor for suicide, with observations indicating a region-specific reduction in the abundance of various astrocyte subsets. For instance, a specific astrocyte subset expressing glutamine synthetase (GLUS), an important enzyme involved in glutamate-to-glutamine salvage to prevent neuroexcitotoxicity, exhibited reduced density in several cortical (dorsolateral prefrontal, subgenual anterior cingulate, and anterior insular), but not subcortical, gray matter regions of depressed subjects compared to healthy controls [26] (Table 1). Furthermore, compared to controls, depressed patients showed a lower abundance of S100B+ astrocytes in the pyramidal layer of the Cornu Ammonis 1 (CA1) region, which serves as the initial component of the hippocampal circuit [27]. Similarly, the abundance of GFAP+ astrocytes was lower in various hippocampal regions—including CA1/2/3 regions and the dentate gyrus—among middle-aged individuals (~50 years of age) with MDD, possibly linked to a decrease in hippocampal volume [28]. Moreover, lower astrocyte density in the hilus region was observed in subjects not taking antidepressants, highlighting the potential influence of this type of medication on astrocyte abnormalities in MDD [28]. Furthermore, alterations in GFAP+ astrocytes in the CA1 region and the dentate gyrus were associated with age and sex, respectively [28]. Interestingly, a different study involving elderly subjects (>60 years of age) with MDD revealed decreased GFAP+ astrocyte counts in the cortical gray matter and increased counts in the cortical white matter, specifically in the prefrontal cortex (PFC) [29]. The discrepancy between these two studies might be explained by the presence of age-related differences in astrocyte pathology in depression. Alternatively, spatial differences in astrocyte distributions in brain specimens from depressed patients could be influenced by the presence of comorbid illnesses, different demographics of study participants, and the use of different tissue sampling methods. Of note, reduced GFAP expression was positively correlated with gamma-aminobutyric acid (GABA) neuron density and negatively correlated with glutamate concentration in the PFC of depressed subjects, providing further evidence for the protective role of astrocytes in preventing glutamate-induced neuroexcitotoxicity in MDD [30]. Collectively, these reductions in astrocyte numbers may indicate an overall decrease in the homeostatic proliferation of these cells. In fact, an astrocyte subset expressing the phosphorylated form of GFAP—which has been implicated in astrocyte mitosis [31]—also showed a reduction in numbers in the PFC of depressed patients [32]. This finding offers a potential mechanistic explanation for the overall reduction in astrocyte abundance observed in the aforementioned studies. It should be noted that the extent of reduction in astrocyte densities may vary among different neuropsychiatric illnesses. For instance, individuals with MDD and bipolar disorder (BD) exhibited a reduction in S100B+ astrocyte abundance, while those with schizophrenia and MDD showed a decline in the abundance of astrocytes expressing phosphorylated GFAP, a feature that is absent in BD [27,28].

Regarding astrocyte phenotypes, transcriptional and immunohistochemical analyses showed that depressed men exhibited a reduction in several astrocyte-related markers in the locus coeruleus and/or hippocampus, including GFAP and high-affinity glutamate transporters (SLC1A1/2/3) [33,34]. The expression of other molecules involved in astrocyte nutrient transport, such as the K+ channel gene 10 (KCNJ10) and Aquaporin 4 (AQP4), was also decreased in the hippocampus of these patients [34]. Furthermore, a reduction in the expression of gap junction-associated molecules, such as connexins 30 (gap junction protein A1, GJA1) and 43 (gap junction protein B6, GJB6), was observed in the hippocampus and orbital frontal cortex of depressed subjects, indicating possible disruptions in astrocyte-mediated barrier function [34,35].

Besides brain tissue-based studies, fluid biomarker analyses revealed various changes in astrocyte-related factors in depressed patients, particularly in the circulating levels of S100B. While there are no reported genetic polymorphisms of S100B in depression [36], this molecule has been linked to the development of depressive symptoms in subjects with comorbid somatic illnesses, such as end-stage renal diseases and myocardial infarction [37,38]. Furthermore, elevated S100B serum levels were linked to depressive symptoms in older healthy adults and individuals experiencing stress, such as burned-out medical residents and combat trainees [39,40,41]. Consistently, elevated serum levels of S100B were observed in MDD patients compared to healthy controls. In a systematic analysis of over 600 patients, increased serum S100B levels were associated with disease severity, female sex, and age [42]. Additionally, S100B serum levels decreased during antidepressive treatment only in subjects with sufficient clinical improvement [43]. Other studies suggested that higher pre-treatment levels of serum S100B could be used to predict the responsiveness to antidepressant treatment [44,45]. Collectively, S100B may serve as a valuable biological indicator with clinical significance for the diagnosis and treatment of depression.

Apart from changes in S100B, MDD patients also exhibited elevated levels of soluble GFAP in the cerebrospinal fluid (CSF) [46], while anti-GFAP autoantibodies were found in depressed patients with psychosis [47]. These findings suggested that dysregulated expression of soluble GFAP might trigger an autoimmune reaction against astrocytes and the subsequent behavioral pathology. Depressed patients also showed increased GFAP levels in the circulation with significant correlations with age and disease severity [48]. Interestingly, depressed subjects did not show any changes in the S100B concentration in the CSF, prompting the possibility of S100B playing different roles in the CNS compared to circulation during MDD development [45]. Lastly, depressed patients have shown elevated levels of extracellular vesicles containing GFAP or those co-expressing AQP4 and GFAP in serum samples, providing additional evidence for the existence of astrocyte dysfunction in depression [49].

It should be noted that discrepancies regarding the alterations of astrocyte-related markers in MDD have been observed. For instance, a recent small-cohort study found no differences in mRNA and protein expression of GFAP in the dorsolateral PFC between age-matched depressed patients and healthy controls, possibly due to the influence of sex and other confounding factors [50]. On a different but related note, a preclinical study did not find an association between GJB6 and depression pathophysiology [51]. This is likely due to the distinct anatomical impact of this molecule on BBB function in various brain regions, as evidenced by the analysis of cerebellar depression in this report compared to the observed defects in the hippocampus and frontal cortex in clinical specimens, and/or species-related differences. Despite these discrepancies, most studies point towards a loss of homeostatic function of astrocytes in MDD, with the involution of selected astrocyte subsets and proteomic and transcriptional alterations of astrocyte-related markers in the brain and circulation (Figure 1).

### 3.2. Astrocyte-Related Abnormality in Suicidal Subjects

Astrocyte-related morphological alterations have been observed in various brain regions of individuals with SI/SB (Figure 1). In this regard, astrocyte hypertrophy, a cardinal feature of reactive astrocytes, was observed in the white matter of the anterior cingulate cortex (ACC) of depressed suicide completers [52] (Table 2). Astrocytes in this brain region also exhibited larger cell bodies with more ramified processes than those in control specimens. These distinct features were also documented in the caudate nucleus and thalamus of suicide completers. Molecular characterization of these astrocytes revealed an increased protein expression of GFAP in the PFC of suicidal subjects compared to controls [53]. Interestingly, a reduced mRNA expression of GFAP was observed in selected brain regions, including the PFC of depressed individuals who died by suicide (depressed suicides) [52,54], along with other astrocyte-related markers, such as aldehyde dehydrogenase 1A1(ALDH1L1), SOX9, glutamate-ammonia ligase (GLUL), and SCL1A3 [55]. These findings suggest that complex transcriptional and translational regulation of GFAP might occur during SI/SB development. On one hand, elevated GFAP protein levels might reflect the hypertrophic state of astrocytes in the brains of individuals experiencing SI/SB. On the other hand, reduced mRNA expression of GFAP and other astrocyte markers might indicate that a molecular pathway underlying the homeostatic identity of these glial cells has been compromised. This hypothesis is supported by the widespread decrease in frequencies of GFAP+ and Vimentin+ astrocyte subsets across different brain regions (particularly, PFC, caudate nucleus, and thalamus) of depressed suicides [56]. Differences in patient categorization (suicide completers vs. depressed suicides) might also contribute to these discrepancies.

Notably, some astrocyte markers might be of clinical utility to distinguish brain samples of suicidal subjects from those with other psychiatric illnesses. In this regard, non-suicidal depressed subjects could be distinguished from depressed suicides by the elevated expression of the astrocyte gene, ALDHL1, in the dorsolateral PFC and ACC regions of the former [57]. Elevated ALDHL1 in the dorsolateral PFC of non-suicidal schizophrenia patients also distinguished them from non-schizophrenic suicide completers [58].

Other abnormal molecular phenotypes associated with suicide have been observed in astrocytes. Specifically, the expression of the truncated variant of the tropomyosin-related kinase B (TrkB.T1) was reduced in the PFC of suicide completers [59]. This reduction was mechanistically linked to a hypermethylation state in the TrkB promoter and an elevated expression of the Homo sapiens microRNA-185 (Hsa-miR-185) [60]. Others suggested that reduced chemokine and immune-related gene expression might be associated with the reduction of astrocyte function in depressed suicides compared to controls [61]. Moreover, suicidal subjects showed abnormalities in molecules involved in astrocyte-dependent BBB integrity. Specifically, the mRNA expression of GJA1 and GJB6 was reduced in the PFC of suicide completers compared to controls, and altered expression of the former was linked to dysregulation of the transcriptional regulator SOX9 [62]. Similarly, hypermethylation-associated downregulation of these two connexins was observed in various brain regions, including neocortex (Brodmann areas 4 and 17, thalamus, and caudate nucleus, but not in the cerebellum (where GJA1 expression was elevated) of depressed suicides [63]. Furthermore, a reduced GJA1 expression in astrocytes was visually confirmed through immunohistochemistry and coincided with the presence of oligodendrocytes in the ACC of these individuals [64]. Collectively, these findings suggest that an altered expression of this gap junction coupling protein might be involved in the abnormal regulation of BBB permeability mediated by glial cells during SI/SB development. In fact, evidence of a “leaky” brain has been reported in the CSF samples of suicidal subjects. For example, hyaluronic acid, the ligand for the putative suicide risk gene CD44, showed elevated levels in the CSF of individuals who attempted suicide, and this increase was positively correlated with the BBB permeability [65]. S100B, an astrocyte-associated marker of BBB disruption, was elevated in the CSF of suicidal subjects compared to individuals who had died of other causes [66]. Serum levels of S100B were also significantly correlated with the severity of SI among adolescent subjects with psychosis or affective disorders [67]. However, another study revealed that elevated serum S100B expression appeared to be an astrocyte-related molecular hallmark across different depression subtypes (psychotic, non-suicidal, and suicidal depression) [57].

In summary, similar to depression, SI/SB was associated with reduced frequencies of selected astrocyte subsets as well as proteomic and transcriptional changes in astrocyte-related markers in a brain-region-specific manner. However, hypertrophic reactive astrocytes appeared to be a unique pathology of SI/SB. These distinct abnormal astrocyte physiologies in depressed vs. suicidal individuals might have implications for astrocyte impairment and subsequent neuroinflammatory processes associated with depression and SI/SB. Mechanistically, astrocyte dysfunction may represent an important cellular pathway underlying the transition from depression to SI/SB. This dysfunction may disrupt neurometabolic homeostasis in vulnerable brain circuits (e.g., hippocampus, PFC, ACC, and other regions previously implicated in SI/SB). Alternatively, astrocyte abnormalities in individuals affected by these two psychiatric conditions might synergistically provide the permissive cues for neuroinflammatory onset and prolongation, thereby heightening the likelihood of developing SI/SB.

It is worth noting that there are methodological caveats (diverse assessment methods of astrocyte phenotypes, heterogeneity in study subjects and tissue types, etc.) in the current literature that prevent a comprehensive comparative analysis of all clinical evidence. Therefore, further research is needed to gain a more comprehensive understanding of these cells in SI, SA, completed suicides, as well as different subtypes of depression, with consideration of comorbid neuropsychiatric and somatic illnesses, as well as medication statuses. Importantly, while astrocytes might represent a trait marker for depression [68], a causal role of astrocytes in SI/SB is difficult to be determined, primarily owing to the absence of preclinical models. Despite these technical challenges, future clinical studies focusing on a detailed characterization of astrocyte dysfunction by measuring circulating astrocyte markers or visualizing in situ astrocyte state using molecular imaging modalities in subjects with SI/SB might help determine whether astrocyte abnormalities represent a state marker of these conditions.

## 4. Astrocyte Dysfunction in COVID-19

Similar to depression and suicide, several astrocyte-related dysfunctions have been observed in the context of SAR-CoV-2 infection. As such, these abnormalities might represent an overlapping pathology in the development of these aforementioned neuropsychiatric conditions and COVID-19. This section provides an overview of the notable phenotypic changes of astrocytes in post-mortem studies of brain specimens from deceased patients with acute COVID-19 as well as the alterations in astrocyte-derived factors from serum samples collected during the acute and long COVID-19 phases.

### 4.1. Astrocyte-Related Abnormality in Acute COVID-19

In brain samples from deceased patients with acute COVID-19, significant alterations in various astrocyte features, including increased GFAP immunoreactivity and astrocytosis, were observed [69,70]. Phenotypically, hypertrophic astrocytes with lower branching complexity and co-expression of GFAP/collagen IV and aquaporin 4 (AQP4) were observed in COVID-19 patients [71]. Astrocytes were also found to be in contact with activated microglia or surrounding dying neurons. During this disease phase, selected circulating markers of reactive astrocytosis were elevated (Table 3). In particular, COVID-19 patients exhibited increased serum S100B levels upon hospital admission compared to healthy controls, showing a positive correlation with disease severity [72,73,74]. Furthermore, among individuals admitted to an intensive care unit (ICU) for COVID-19, serum S100B levels were elevated in non-survivors compared to survivors, showing a notable correlation with inflammatory markers, such as lymphocyte counts and interleukin-6 (IL-6) levels [75]. Besides S100B, an increased expression of serum GFAP, sampled during the acute disease phase or at hospital admission [76,77], was accompanied by a positive correlation with disease severity and the occurrence of critical polyneuropathy/myopathy illness [78].

It is worth noting that discrepancies exist regarding S100B and GFAP expression in acute COVID-19. For example, serum S100B sampled at an unspecified time during the acute disease phase showed no significant difference compared to controls [76], which might be explained by differences in sample collection schedule (at hospital admission vs. anytime during the acute disease phase). Furthermore, the association between GFAP and COVID-19-related neurological symptoms also represents a matter of debate, which might be related to the characteristics of neurological manifestations (onset time and numbers/types of symptoms) and/or sampling time. In this regard, elevated GFAP levels were reportedly not associated with the presence of neurological symptoms [76]. In a different study, higher levels of the serum neuronal injury marker Ubiquitin carboxy-terminal hydrolase L1 (UCHL1), but not GFAP, were linked to neurological manifestations in COVID-19 patients at ICU admission [79]. Furthermore, two other studies demonstrated that serum GFAP was elevated in COVID-19 patients with no neurological symptoms compared to controls [80] and normalized over the time course of 2 months after recovery [81]. However, in an analysis comparing severe COVID-19 patients to those with neuro-COVID-19 (characterized by neurological complications detected upon hospitalization and confirmed through electrophysical and other assessment metrics), serum GFAP levels were elevated in both cohorts compared to controls [82]. Evidence for the absence of elevated serum GFAP sampled at admission time in COVID-19 patients also exists [83], which might be attributable to the heterogeneity of disease severity among different COVID-19 patient cohorts.

### 4.2. Astrocyte-Related Abnormality in Long COVID-19

To date, studies on astrocyte-related biomarkers in subjects with long COVID-19 have mostly focused on the assessment of serum GFAP levels and produced conflicting outcomes (Table 3). In a comparative analysis between subjects with severe COVID-19 and those with mild long COVID-19 (characterized by persistent headache for more than 12 weeks after initial serological sampling), serum GFAP levels were significantly higher in the former cohort [84]. Additionally, a longitudinal analysis of serum GFAP In COVID-19 patients with varying severity revealed a normalization of GFAP expression 6 months after the initial serological analysis, despite the persistence of neurological symptoms in 50% of the cohort (i.e., fatigue, brain fog, and impaired cognition) [85]. In contrast, the elevated serum GFAP in hospitalized COVID-19 patients was reportedly associated with long-term neurological symptoms (monitored up to 1 year after hospital admission) [86]. Furthermore, in subjects with long COVID-19 and post-acute neurological complications, anxiety and depression were significantly associated with higher serum glial reactivity, marked by elevated neuroglial GFAP and Neurofilament Light Polypeptide (NFL) scores [87]. Discrepancies in the association between reactive astrocytosis and neurological complications in long COVID-19 might be attributed to serum kinetics of these astrocyte markers after infection. Indeed, long COVID-19 subjects with post-acute neurological complications showed elevated serum GFAP during early (<90–120 days), but not late, recovery [88]. Alternatively, these findings might suggest that reactive astrocytosis was associated with certain neurological manifestations in long COVID-19 patients (i.e., depression), but not with other types of symptoms (e.g., fatigue, headache, etc.).

In summary, the most convincing evidence for the presence of astrocyte dysfunction in COVID-19 stems from the few existing post-mortem studies, which suggested the transition from a homeostatic to an activated/reactive phenotype of these glial cells during the acute disease phase. In contrast, the absence of methodological standardization (chiefly related to patient stratification and tissue sampling time) has produced some inconsistencies in serum biomarker studies. Nevertheless, many reports pointed towards an increase in the circulating markers of astrocyte activation, namely S100B and GFAP, during the acute COVID-19 phase. Regarding long COVID-19, we anticipate that the recent development of a unified consensus on the diagnosis and evaluation of symptoms associated with this condition will allow a more comprehensive and accurate characterization of the potential role of astrocyte dysfunction in this emerging illness of public health concern [89].

## 5. Reactive Astrocytosis as a Precipitating Factor for SI/SB Development in Long COVID-19?

The aforementioned in situ findings of astrocyte dysfunction in COVID-19 suggest the presence of these cells in a neuro-immune communication hub, where they might orchestrate hyper-inflammatory pathology in COVID-19. These observations also indicate that astrocyte dysfunction might represent a shared pathology between COVID-19 and SI/SB development. In fact, the transcriptional analysis of brain samples from COVID-19 patients revealed marked changes in astrocyte signaling pathways, resembling those of the astrocytes derived from patients with neurological illnesses [90]. Hence, these astrocyte pathologies might underlie the development of neurological sequelae, including SB/SI, in long COVID-19. Some important mechanistic insights regarding this putative pathological pathway from astrocyte dysfunction to SI/SB development in COVID-19 have emerged in more recent literature.

Firstly, long-term CNS infection with SARS-CoV-2 might occur during COVID-19 pathogenesis [91], transforming homeostatic astrocytes into a reactive state. Furthermore, both SARS-CoV-2 and elevated GFAP were reportedly detectable in the CSF of a COVID-19 patient nearly three weeks after symptom onset and after having tested negative twice for the virus [92]. Collectively, these observations support the potential presence of a long-term infected and chronically reactive state of astrocytes in COVID-19.

Secondly, chronic reactive astrocytosis might be triggered and maintained by the direct infection of these glial cells by SARS-CoV-2. This hypothesis is supported by the data from a study showing that the majority of SARS-CoV-2 spike-containing neural cells in a subset of COVID-19 patients were identified as astrocytes [93]. In fact, recent studies revealed that both developmental and adult astrocytes were highly susceptible cellular targets of SARS-CoV-2 through an ACE2-independent entry mechanism. Mechanistically, SARS-CoV-2 may utilize dipeptidyl peptidase 4 (DPP4)—the main receptor of the Middle East respiratory syndrome-related coronavirus (MERS-CoV)—to interact with its receptor basigin (BSG/CD147) on astrocytes [94]. This interaction occurs within a cellular transmission pathway of SARS-CoV-2, facilitating the transmission from infected astrocytes to pericytes and, subsequently, to the nearby non-infected astrocyte endfeet [94]. Ultimately, this pathway might prolong CNS infection of these glial cells and sustain their dysfunction.

Thirdly, chronically reactive astrocytes might lose their ability to carry out various neurotrophic support functions that they perform in their homeostatic state, including BBB regulation. Consequently, a compromised CNS barrier promotes further viral invasion and trafficking of peripheral inflammatory mediators into the brain, leading to a hyper-inflammatory state. Of note, recent studies suggested the presence of a disrupted BBB in individuals with long COVID-19. In particular, astrocyte perturbations occurred with abnormalities in junctional proteins within the vascular basement membranes [95], suggesting that a compromised BBB might result from reactive astrocytosis. Furthermore, a longitudinal analysis of serum matrix metalloproteinase-9 (MMP-9) levels revealed a significant and persistent increase (up to 2 weeks after recovery) in this marker of BBB disruption among COVID-19 patients with neurological sequelae, compared to severe COVID-19 patients and healthy controls [82]. Consistent with this finding, an extended biochemical and imaging analysis of severe neuro-COVID-19 patients (up to 13 months after recovery) showed that neurological complications from COVID-19 infection were accompanied by BBB disruption and increased neuroinflammation [96]. Taken together, these findings suggest the possible existence of a neuroinflammatory circuit driven by chronically reactive astrocyte-mediated disruption of BBB integrity in long COVID-19 patients with neurological sequelae.

Fourthly, chronic hyperinflammation resulting from the loss of astrocyte support for BBB integrity might increase suicide prevalence in long COVID-19 patients (Figure 2). In this context, inflammation has been implicated as a shared pathology between suicide and depression [10,97]. Mechanistically, the CNS inflammatory milieu might trigger serotonergic and glutamatergic neurotoxicity [98] and/or excitoxicity in vulnerable brain circuits that have been implicated in SI/SB development. Furthermore, in the context of long COVID-19, such molecular and cellular pathology of neuroinflammation might explain the potential longitudinal overlap between chronic reactive astrocytosis, leaky BBB, and the delayed rise in suicide prevalence after COVID-19 infection [99]. Notably, the observed association between increased suicide prevalence and neuropsychiatric symptoms has been recently documented in mid- to long-term studies of long COVID-19 patients, as detailed below.

A French study of 115 long COVID-19 patients revealed a significant increase in suicide prevalence at 4 months after ICU admission [100]. Moreover, a cohort study of 506 American adults with a past COVID-19 diagnosis showed significantly higher SI and SB scores up to one year after infection [101]. Furthermore, some neuropsychiatric risk factors of SB, such as MDD, were recently linked to a subset of long COVID-19 patients [102,103], suggesting that long COVID-19-associated depression might increase SB. In this regard, mid- to long-term studies (6–9 months) of long COVID-19 patients in France (*n* = 60) and Brazil (*n* = 425) revealed a prevalence of depression exceeding 30% in the French study [102] and 8% in the Brazilian study [103]. In addition, depressive symptoms reportedly persisted beyond 6 months in COVID-19 long-haulers [104], with possible association to female sex [105]. These observations were further supported by a clinical observation covering six nations, which revealed an increased risk of depression at 16 months after COVID-19 infection among over 240,000 patients [106], and preclinical findings of depressive behaviors in mice triggered by COVID-19/SARS-CoV-2 infections [107,108]. However, it is worth noting that depressed subjects reportedly did not exhibit increased susceptibility to COVID-19 infection [109,110]. Besides depression, reduced self-control was also observed in long COVID-19 patients, prompting the possibility of an increase in impulsivity, another risk factor for suicide, in these subjects [111]. Collectively, these findings suggest the potential existence of a pathological mechanism through which SARS-CoV-2 hijacks astrocytes during neuroinvasion, thereby rendering the brain permeable to inflammatory infiltrates even during the post-acute phases of COVID-19. This pathway of astrocyte-mediated neuroinflammation might lead to an increased SI/SB prevalence, along with other neuropsychiatric sequelae in long COVID-19 (Figure 2).

Of important mechanistic consideration, the potential increase in suicide among long COVID-19 patients might not be specific to SARS-CoV-2 infection. Firstly, several pathogens, ranging from *Toxoplasma gondii* and H1N1 to other coronaviruses (MERS-CoV and SARS-CoV-1), have been linked to increased depression and/or suicide in geographically distinct populations [112,113,114,115,116]. While a consensus among these association studies has not been reached [117,118,119], pathogen-induced BBB leakage might occur in different types of infectious illnesses with documented neurotropism, thus accounting for the possible increase in depression and suicide among the infected. Future imaging analysis of BBB integrity in patients experiencing neurological symptoms from these types of infections is warranted to confirm this hypothesis. Secondly, prototypical neuroinflammation, emanating from other cellular sources rather than reactive astrocytes, might represent the primary driver of depression and SI/SB. In fact, depression and suicide have been linked to other chronic neurobiological conditions [120,121], which share this pathological feature of neuroinflammation with long COVID-19. Lastly, increased depression and suicide in long COVID-19 might result from dysregulation of pathways other than those of inflammatory origins. In support of this alternative view, no genetic predispositions, including those affecting inflammatory signaling and the immune system, have been linked to depression and suicide in the context of COVID-19 [122]. Furthermore, lower levels of serotonin have recently been linked to long COVID-19 [123], prompting the possibility that defective serotonergic transmission might represent an underlying mechanism of depression and SI/SB [124] in the context of this infectious illness.

## 6. Conclusions

Astrocyte dysfunction has emerged as a potential feature that is shared between depression and SI/SB as well as COVID-19. While reactive astrocytosis and its potential consequence of BBB disruption/hyperinflammation might contribute to suicide development in long COVID-19 patients, the precise mechanism and the relative contribution of astrocyte dysfunction to this chronic neuropsychiatric sequela of SARS-CoV-2 infection remain to be empirically determined. For instance, long-term, inclusive, and comprehensive follow ups of long COVID-19 patients, which are lacking in current literature, may allow further investigation of the possible coexistence of reactive astrocyte pathology with BBB leakage as well as clinical features of SI/SB. Furthermore, such well-controlled longitudinal analyses may reveal whether astrocyte dysfunction represents a novel risk factor of depression and SI/SB in the context of long COVID-19. Lastly, given the proposed mechanistic explanation for the possible rise in SI/SB in the context of long COVID-19, it is important to consider therapeutic strategies for restoring astrocyte homeostasis and enhancing BBB integrity to curb neuroinflammatory and neuropsychiatric manifestations associated with SI/SB among the affected patients.

## Figures and Tables

**Figure 1 brainsci-14-00973-f001:**
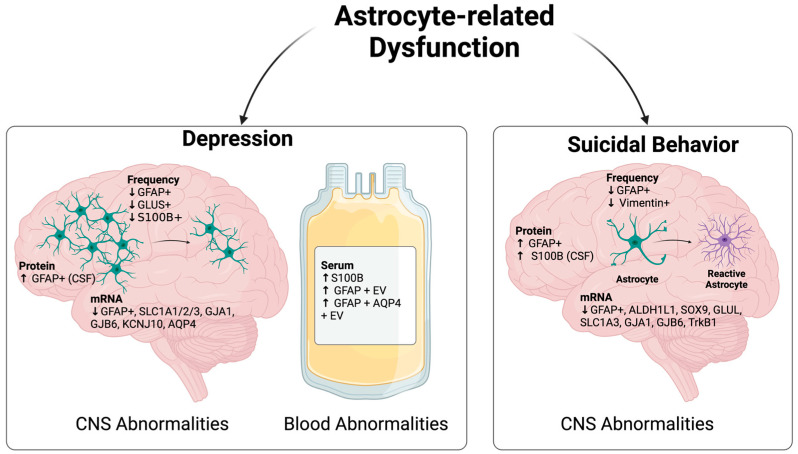
Astrocyte perturbations in individuals affected by depression or suicidal behavior/suicidal ideation. Changes in astrocyte abundance and astrocyte-related gene/protein expression were observed in both depressed subjects (**left**) and those with suicidal ideation/suicidal behavior (SI/SB) (**right**). Furthermore, depressed patients showed alterations in circulating levels of astrocyte-related markers, while subjects with SI/SB experienced reactive astrocytosis. Abbreviations: ALDH1L1: Aldehyde Dehydrogenase 1 Family Member L1; AQP4: Aquaporin 4; CNS: Central Nervous System; CSF: Cerebrospinal Fluid; EV: Extracellular Vesicles; GFAP: Glial Fibrillary Acidic Protein; GJA1: Gap Junction Protein Alpha 1; GJB6: Gap Junction Protein Beta 6; GLUL: Glutamate-Ammonia Ligase; GLUS: Glutamine Synthetase; KCNJ10: Potassium Inwardly Rectifying Channel Subfamily J Member 10; S100B: S100 Calcium-Binding Protein B; SLC1A1/2/3: Solute Carrier Family 1 Member 1/2/3; SOX9: SRY-Box Transcription Factor 9; TrkB1: Tropomyosin Receptor Kinase B1.

**Figure 2 brainsci-14-00973-f002:**
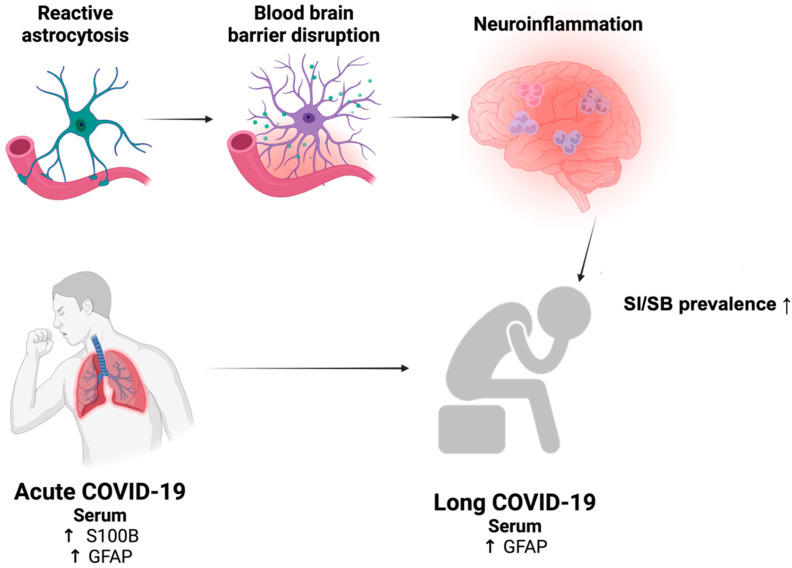
Astrocyte dysfunction and their possible involvement in suicide development during COVID-19 pathogenesis. COVID-19 is characterized by various astrocyte-related dysfunctions, including reactive astrocytosis and potential changes in the circulating astrocyte-related markers (S100B and/or GFAP) in different phases of COVID-19 (acute vs. long). Importantly, these astrocyte abnormalities might represent an overlapping pathology between suicide and long COVID-19 via chronic reactive astrocytosis-mediated disruption of the blood–brain barrier, which promotes neuroinflammation and subsequently heightens the prevalence of SI/SB.

**Table 1 brainsci-14-00973-t001:** Details of clinical studies of astrocyte perturbations in individuals affected by depression.

	Study Type and Tissue	Study Methods	Study Participants	Major Findings
Bernstein et al. [26]	Post-mortem brain biopsy	Immunohistochemistry	MDD (*n* = 14) vs. HC (*n* = 16)	Reduced GLUS+ astrocytes in DLPFC, sACC, and AiC in MDD compared to HC
Gos et al. [27]	Post-mortem brain biopsy	Immunohistochemistry	MDD (*n* = 9) vs. HC (*n* = 13)	Reduced S100B+ astrocytes in pyramidal CA1 in MDD compared to HC
Cobb et al. [28]	Post-mortem brain biopsy	Immunohistochemistry	MDD (*n* = 17) vs. HC (*n* = 17)	Reduced GFAP+ astrocytes in hippocampal CA1/2/3 and dentate gyrus in MDD compared to HC
Davis et al. [29]	Post-mortem brain biopsy	Immunohistochemistry	MDD (*n* = 20) vs. HC (*n* = 20)	Reduced GFAP+ astrocytes in cortical gray matter and increased GFAP+ astrocytes in PFC white matter in MDD compared to HC
Oh et al. [30]	Post-mortem brain biopsy	Immunohistochemistry	MDD (*n* = 15) vs. HC (*n* = 15)	Reduced GFAP correlated positively with GABA+ neuron density and negatively with glutamate concentration in PFC of MDD patients.
Webster et al. [32]	Post-mortem brain biopsy	Immunohistochemistry	MDD (*n* = 15) vs. HC (*n* = 15)	Reduced phosphorylated GFAP+ astrocytes in PFC in MDD compared to HC
Chandley et al. [33]	Post-mortem brain biopsy	Immunohistochemistry, gene expression	MDD (*n* = 19) vs. HC (*n* = 20)	Reduced GFAP and SLCA1/2/3 mRNA as well as GFAP protein in locus coeruleus and/or hippocampus in MDD compared to HC
Medina et al. [34]	Post-mortem brain biopsy	Gene expression	MDD (*n* = 13) vs. HC (*n* = 10)	Decreased KCNJ10, AQP4, GJA1, and SLC1A2/3 mRNA in hippocampus in MDD compared to HC
Miguel-Hidalgo et al. [35]	Post-mortem brain biopsy	Immunohistochemistry	MDD (*n* = 23) vs. HC (*n* = 20)	Decreased GJA1 and GJB6 protein in OFC in MDD compared to HC
Yang et al. [36]	Blood	Genetic polymorphism	MDD (*n* = 152) vs. HC (*n* = 150)	No association between S100B polymorphism and MDD
Kim et al. [37]	Serum	ELISA	End-stage renal disease (*n* = 78)	Higher S100B was linked to depressive symptoms.
Tulner et al. [38]	Serum	ELISA	Myocardial infarction (*n* = 48)	Higher S100B was linked to depressive symptoms.
Benitez et al. [39]	Serum	ELISA	Elderly HC (*n* = 35)	Higher S100B was linked to depressive symptoms.
Gulen et al. [40]	Serum	ELISA	Burned-out medical residents (*n* = 48)	Higher S100B was linked to depressive symptoms.
Li et al. [41]	Serum	ELISA	Combat trainees (*n* = 37)	Higher S100B was linked to depressive symptoms.
Tural et al. [42]	Serum	Meta-analysis	MDD (*n* = 658)	Higher S100B was linked to depression severity, age of MDD onset, and female sex.
Schroeter et al. [43]	Serum	ELISA	MDD (*n* = 193) vs. HC (*n* = 131)	Higher S100B in MDD compared to HC; S100B levels decrease after antidepressant treatment correlated with clinical improvement.
Arolt et al. [44]	Serum	ELISA	MDD (*n* = 25) vs. HC (*n* = 25)	Higher S100B in MDD compared to HC; S100B levels correlated positively with treatment response.
Navines et al. [45]	Serum	ELISA	MDD (*n* = 27)	S100B levels correlated positively with treatment response.
Michel et al. [46]	CSF	ELISA	Unipolar depression (*n* = 102) vs. HC (*n* = 39)	Higher GFAP in patients compared to HC
Endres et al. [47]	CSF	ELISA	MDD with psychosis (*n* = 2)	Higher GFAP autoantibodies in patients compared to HC
Steinacker et al. [48]	Serum	ELISA	MDD (*n* = 81) vs. HC (*n* = 81)	Higher GFAP in MDD compared to HC; GFAP levels positively correlated with disease severity.
Wallensten et al. [49]	Plasma	ELISA	MDD (*n* = 31) vs. HC (*n* = 61)	Increased extracellular vesicles co-expressing AQP4/GFAP in MDD compared to HC
Qi et al. [50]	Post-mortem brain biopsy	Immunohistochemistry, gene expression	MDD (*n* = 5) vs. HC (*n* = 12)	No differences in GFAP mRNA and protein expression in DLPFC of MDD compared to HC

Abbreviations: AiC: Anterior insular Cortex; AQP4: Aquaporin 4; CA1: Cornu Ammonis 1; CSF: Cerebrospinal Fluid; DLPFC: Dorsolateral Prefrontal Cortex; ELISA: enzyme-linked immunosorbent assay; GABA: Gamma-Aminobutyric Acid; GFAP: Glial Fibrillary Acidic Protein; GJA1: Gap Junction Protein Alpha 1; GJB6: Gap Junction Protein Beta 6; GLUS: Glutamine Synthetase; HC: Healthy Control; KCNJ10: Potassium Inwardly Rectifying Channel Subfamily J Member 10; MDD: Major Depressive Disorder; OFC: Orbitofrontal Cortex; S100B: S100 Calcium-Binding Protein B; sACC: subgenual Anterior Cingulate Cortex; and SLC1A1/2/3: high-affinity glutamate transporters 1A1/2/3.

**Table 2 brainsci-14-00973-t002:** Details of clinical studies of astrocyte perturbations in individuals affected by suicide.

	Study Type and Tissue	Study Methods	Study Participants	Major Findings
Torres-Platas et al. [52]	Post-mortem brain biopsy	Immunohistochemistry	Depressed suicide (*n* = 10) vs. Sudden-death control (*n* = 10)	Astrocyte hypertrophy with increased cell body and ramifications in ACC white matter, caudate nucleus, and thalamus of depressed suicides compared to controls
Schlicht et al. [53]	Post-mortem brain biopsy	Mass spectrometry	Suicide (*n* = 17) vs. Accident or heart-disease control (*n* = 9)	Increased GFAP in PFC of suicide completers compared to controls
Torres-Platas et al. 2015 [54]	Post-mortem brain biopsy	Immunohistochemistry, gene expression	Depressed suicide (*n* = 22) vs. Accident or sudden-death control (*n* = 22)	GFAP mRNA and protein were reduced in mediodorsal thalamus and caudate nucleus of depressed suicides compared to controls.
Nagy et al. [55]	Post-mortem brain biopsy	Gene expression (methylation pattern)	Suicide (*n* = 22) vs. Sudden-death control (*n* = 17)	Decreased expression of GFAP, ALDH1L1, SLC1A3, GJA1, GJB6, GLUL, and SOX9 in DLPFC of suicides compared to controls
O’Leary et al. [56]	Post-mortem brain biopsy	Immunohistochemistry	Depressed suicide (*n* = 10) vs. Sudden-death control (*n* = 10)	Decreased abundance of Vimentin+ and GFAP+ astrocytes in DLPFC, dorsal caudate nucleus, and mediodorsal thalamus of depressed suicides compared to controls
Zhang et al. [57]	Post-mortem brain biopsy	Gene expression	Depressed suicide (*n* = 17) vs. Non-suicidal depressed (*n* = 17)	Decreased ALDHL1 in the DLPFC and ACC regions of depressed suicides compared to non-suicidal depressed
Zhang et al. [58]	Post-mortem brain biopsy	Gene expression	Non-schizophrenic suicide (*n* = 7) vs. non-suicidal schizophrenic (*n* = 28)	Decreased ALDHL1 in the DLPFC of non-schizophrenic suicides compared to non-suicidal schizophrenics
Maussion et al. [59]	Post-mortem brain biopsy	Gene expression (methylation pattern)	Suicide (*n* = 13) vs. Control (*n* = 11)	Decreased TrkB.T1 expression in the frontal cortex of suicides compared to controls
Maussion et al. [60]	Post-mortem brain biopsy	Gene expression (miRNA profile)	Suicide (*n* = 38) vs. Control (*n* = 17)	Increased Hsa-miR-185 expression in the frontal cortex of suicides compared to controls
Pantazatos et al. [61]	Post-mortem brain biopsy	Gene expression	Depressed suicide (*n* = 21) vs. Sudden-death control (*n* = 29)	Lower expression of genes associated with astrocyte migration in DLPFC of depressed suicides compared to controls
Ernst et al. [62]	Post-mortem brain biopsy	Gene expression	Suicide (*n* = 95) vs. Sudden-death control (*n* = 81)	Decreased expression of GJA1 and GJB6 in DLPFC of depressed suicides compared to controls
Nagy et al. [63]	Post-mortem brain biopsy	Gene expression (methylation pattern)	Depressed suicide (*n* = 22) vs. Sudden-death control (*n* = 22)	Decreased expression of GJA1 and GJB6 as well as increased H3K9me3 in these two gene loci in the neocortex (Brodmann areas 4 and 17), mediodorsal thalamus, and caudate of depressed suicides compared to controls
Tanti et al. [64]	Post-mortem brain biopsy	Immunohistochemistry	Depressed suicide (*n* = 48) vs. Sudden-death control (*n* = 23)	Decreased GJA1 in astrocytes adjacent to oligodendrocytes in ACC of depressed suicides compared to controls
Dogan et al. [65]	Post-mortem CSF	ELISA	Suicide (*n* = 32) vs. Heart-disease or other-cause control (*n* = 56)	Higher S100B in suicides compared to controls
Falcone et al. [66]	Serum	ELISA	Suicide with psychosis (*n* = 40) or affective disorders (*n* = 24) vs. Control (*n* = 20)	Increased S100B correlates positively with SI severity in suicides compared to controls.

Abbreviations: ACC: Anterior Cingulate Cortex; ALDH1L1: Aldehyde Dehydrogenase 1A1; CSF: Cerebrospinal Fluid; DLPFC: Dorsolateral Prefrontal Cortex; ELISA: enzyme-linked immunosorbent assay; GFAP: Glial Fibrillary Acidic Protein; GJA1: Gap Junction Protein Alpha 1; GJB6: Gap Junction Protein Beta 6; GLUL: glutamate-ammonia ligase; S100B: S100 Calcium-Binding Protein B; SLC1A3: high-affinity glutamate transporter 1 A3; TrkB.T1: truncated variant of the tropomyosin-related kinase B.

**Table 3 brainsci-14-00973-t003:** Details of clinical studies of astrocyte perturbations in individuals affected by acute vs. long COVID-19.

	Study Type and Tissue	Study Methods	Study Participants	Major Findings
Acute COVID-19				
Boroujeni et al. [69]	Post-mortem brain biopsy	Immunohistochemistry	Acute COVID-19 (*n* = 3) vs. HC (*n* = 3)	Increased astrocyte abundance in cerebral cortex of COVID-19 patients compared to controls
Reichard et al. [70]	Post-mortem brain biopsy	Immunohistochemistry	Acute COVID-19 with disseminated encephalomyelitis (*n* = 1)	Increased GFAP expression in cerebral white matter in a COVID-19 patient
Lee et al. [71]	Post-mortem brain biopsy	Immunohistochemistry	Acute COVID-19 (*n* = 13)	Hypertrophic astrocytes with lower branching complexity and GFAP/collagen IV and AQP4 co-localization in close proximity to microglia/dying neurons in COVID-19 patients
Aceti et al. [72]	Serum	ELISA	Acute COVID-19 (*n* = 74) vs. HC (*n* = 5)	Increased S100B in COVID-19 compared to controls upon hospitalization; S100B correlated positively with inflammation, organ damage markers, and disease severity.
Mete et al. [73]	Serum	ELISA	Acute COVID-19 (*n* = 64) vs. HC (*n* = 30)	Increased S100B in COVID-19 compared to controls upon hospitalization; S100B correlated positively with disease severity.
Silva et al. [74]	Serum	ELISA	Acute COVID-19 (*n* = 141) vs. HC (*n* = 36)	Increased S100B in COVID-19 compared to controls at 14 days after disease onset; S100B correlated positively with disease severity.
Kokkoris et al. [75]	Serum	ELISA	Acute COVID-19 (*n* = 50)	Higher S100B in non-survivors compared to survivors among ICU patients upon hospitalization; S100B positively correlated with IL-6, low lymphocyte count, hypoperfusion indices, disease severity, and short-term outcome.
Sahin et al. [76]	Serum	ELISA	Acute COVID-19 (*n* = 58) vs. HC (*n* = 20)	No difference in S100B between two groups; higher GFAP in severe COVID-19 compared to controls (unspecified time during acute disease).
Passos et al. [77]	Serum	ELISA	Acute COVID-19 (*n* = 42) vs. HC (*n* = 34)	Higher GFAP in severe COVID-19 compared to controls upon hospitalization; GFAP positively correlated with RAGE and HMGB1 levels.
Frithiof et al. [78]	Serum	ELISA	Acute COVID-19 (*n* = 111)	Higher GFAP correlated with polyneuropathy and myopathy (unspecified time during acute disease).
Tokic et al. [79]	Serum	ELISA	Acute COVID-19 (*n* = 65)	Higher UCHL1, not GFAP, correlated with the presence of neurological symptoms upon admission in male patients.
Plantone et al. [80]	Serum	ELISA	Acute COVID-19 (*n* = 148) vs. HC (*n* = 108)	Higher GFAP in COVID-19 compared to controls (unspecified time during acute disease)
Lennol et al. [81]	Serum	ELISA	Acute COVID-19 (*n* = 45)	Higher GFAP correlated with the absence of neurological symptoms; GFAP normalized during recovery (2 months afterwards).
Bonetto et al. [82]	Serum	ELISA	Acute COVID-19 (*n* = 157) vs. HC (*n* = 20)	Higher GFAP in COVID-19 compared to controls (unspecified time during acute disease) regardless of the presence of neurological symptoms
Savarraj et al. [83]	Serum	ELISA	Acute COVID-19 (*n* = 57) vs. HC (*n* = 20)	No differences in GFAP between two groups upon hospitalization
Long COVID-19				
de Boni et al. [84]	Serum	ELISA	Long COVID-19 (*n* = 6) vs. Severe Acute COVID-19 (*n* = 11)	Reduced GFAP in long COVID-19 (>12 weeks of headache) compared to severe acute COVID-19
Kanberg et al. [85]	Serum	ELISA	Long COVID-19 (*n* = 50) vs. Acute COVID-19 (*n* = 50)	GFAP normalized after 6 months despite persistent long COVID-19 symptoms (fatigue, brain fog, and impaired cognition).
Spanos et al. [86]	Serum	ELISA	Long COVID-19 (*n* = 21) vs. Acute COVID-19 (*n* = 32)	Higher GFAP was linked to long COVID-19 (neurological complications at 1-year follow-up).
Hanson et al. [87]	Serum	ELISA	Long COVID-19 (*n* = 47) vs. Severe Acute COVID-19 (*n* = 9)	Anxiety and depression correlated positively with higher serum neuroglial GFAP in long COVID-19 (post-acute neurological complications).
Peluso et al. [88]	Serum	ELISA	Long COVID-19 (*n* = 52) vs. Acute COVID-19 (*n* = 69)	Higher GFAP during early (<90–120 days), but not late, recovery in long COVID-19 compared to acute COVID-19

Abbreviations: AQP4: Aquaporin 4; ELISA: enzyme-linked immunosorbent assay; GFAP: Glial Fibrillary Acidic Protein; HC: Healthy Control; HMGB1: high mobility group box-1 protein; ICU: Intensive Care Unit; RAGE: receptor for advanced glycation end products; S100B: S100 Calcium-Binding Protein B; UCHL1: Ubiquitin carboxy-terminal hydrolase L1.

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
