# Peer review of "Reactive Astrocytosis—A Potential Contributor to Increased Suicide in Long COVID-19 Patients?"

_brainsci, 2024, doi:10.3390/brainsci14100973_

Round 1

Reviewer 1 Report

Comments and Suggestions for Authors

This is a well-written paper. I have only a few suggestions for consideration:

1.      Abstract results – actual findings of the review need to be presented

2.      Are the astrocyte abnormalities a trait or state marker in depression or SI/SB? This may be clarified.

3.      When authors say risk factor, is there sufficient evidence from longitudinal studies? If not this and some other key limitations in the literature may be highlighted.

Comments on the Quality of English Language

Acceptable.

Author Response

We thank the reviewers for their constructive comments, which have helped to improve our manuscript. Please find below our point-to-point responses to all comments. Revisions to the original manuscript have also been included as red-font texts. We are looking forward to a positive response from the reviewers and editors of Brain Sciences

Reviewer 1

1.Abstract results – actual findings of the review need to be presented

Thank you for this constructive comment. A revised version of the abstract with a summary of actual findings has been supplemented. We also removed the graphical abstract which represented mostly hypothetical details

“Abstract: Background: Long COVID-19 is an emerging chronic illness of significant public health concern due to a myriad of neuropsychiatric sequelae, including increased suicidal ideation (SI) and behavior (SB). Methods: This review provides a concise synthesis of clinical evidence that points toward dysfunction of astrocytes, the most abundant glial cell type in the central nervous system, as a potential shared pathology between SI/SB and COVID-19. Results: Depression, a suicide risk factor, and SI/SB were both associated with reduced frequencies of various astrocyte subsets and complex proteomic/transcriptional changes of astrocyte-related markers in a brain region-specific manner. Astrocyte-related circulating markers were increased in depressed subjects and, to a less consistent extent, in COVID-19 patients. Furthermore, reactive astrocytosis was observed in subjects with SI/SB and those with COVID-19. Conclusions: Astrocyte dysfunctions occurred in depression, SI/SB, and COVID-19. Reactive astrocyte-mediated loss of the blood-brain barrier (BBB) integrity loss and subsequent neuroinflammation —a factor previously linked to SI/SB development— might contribute to increased suicide in individuals with long COVID-19. As such, formulation of new therapeutic strategies to restore astrocyte homeostasis, enhance BBB integrity, and mitigate neuroinflammation, may reduce SI/SB-associated neuropsychiatric manifestations among long COVID-19 patients.”

2.Are the astrocyte abnormalities a trait or state marker in depression or SI/SB? This may be clarified.

Thank you for this thought-provoking comment. While astrocytes might represent a trait marker for depression (Lu et al., Biol Psychiatry 2024), their causal role in SI/SB is difficult to be determined, primarily owing to the absence of preclinical models of this neuropsychiatric condition. Despite these technical challenges, correlation analyses of astrocyte activation phenotype with the clinical severity of SI/SB or depression could be conducted. Future clinical studies with a focus on detailed characterization of astrocyte dysfunction, possibly by measurement of circulating astrocyte markers or visualization of in situ astrocyte functional state by molecular imaging modalities, might determine whether astrocyte abnormalities represent a state marker of these conditions. This issue has been explained in further details at the end of the sections describing astrocyte abnormalities in depression and SI/SB.

“Importantly, while astrocytes might represent a trait marker for depression [68], a causal role of astrocytes in SI/SB is difficult to be determined, primarily owing to the absence of preclinical models. Despite these technical challenges, future clinical studies with a focus on detailed characterization of astrocyte dysfunction, by means of measuring circulating astrocyte markers or visualizing in situ astrocyte state by molecular imaging modalities in subjects with SI/SB, might help determine whether astrocyte abnormalities represent a state marker of these conditions.”

  1. When authors say risk factor, is there sufficient evidence from longitudinal studies? If not this and some other key limitations in the literature may be highlighted.

Thank you for this excellent point! We have reviewed all cited clinical findings and noted that only in reference 106 (Magnúsdóttir et al., Lancet Public Health 2022), depression risk was demonstrably elevated in long COVID-19 patients. In all other works, there was not sufficient evidence for astrocyte dysfunction/inflammation as a risk factor for SI/SB (outside or within the context of COVID-19). As such, we have revised the term “suicide risk” to “suicide prevalence” as astrocyte dysfunction and/or hyperinflammation was associated with increased prevalence of SI/SB in the cited studies. Furthermore, we pointed out the caveats of the current studies of astrocyte abnormalities in depression, SI/SB, and COVID-19 at the end of these sections.

“It's worth noting that there are methodological caveats (diverse assessment methods of astrocyte phenotypes, heterogeneity in study subjects and tissue types, etc.) in the current literature that prevent a throughout comparative analysis of all clinical evidence. Therefore, further research is needed to gain a more comprehensive understanding of these cells in SI, SA, completed suicides, as well as different subtypes of depression, with consideration of comorbid neuropsychiatric and somatic illnesses, as well as medication statuses.”

“…the absence of methodological standardization (chiefly related to patient stratification and tissue sampling time) has produced some inconsistencies in serum biomarker studies. Nevertheless, many reports pointed toward an increase in circulating markers of astrocyte activation, namely S100B and GFAP, during acute COVID-19 phase. Regarding long COVID-19, we anticipate that the recent development of a unified consensus on the diagnosis and evaluation of symptoms associated with this condition will allow a more comprehensive and accurate characterization of the potential role of astrocyte dysfunction in this emerging illness of public health concern [89].”

Finally, we acknowledge the absence of well-designed longitudinal studies to study molecular mechanisms and risk factors of SI/SB in the context of long COVID-19 and emphasize that that such studies are warranted to explore this important topic in the concluding remarks of the review.

“While reactive astrocytosis and its potential consequence of BBB disrup-tion/hyperinflammation might contribute to suicide development in long COVID-19, the precise mechanism and the relative contribution of astrocyte dysfunction to this chronic neuropsychiatric sequela of SARS-CoV-2 infection remains to be empirically determined. For instance, long-term, inclusive, and comprehensive follow-ups of long COVID-19 pa-tients, which are lacking in current literature, may allow further investigation of the possible coexistence of reactive astrocyte pathology with BBB leakage as well as clinical features of SI/SB. Furthermore, such well-controlled longitudinal analyses may reveal whether astrocyte dysfunction represents a novel risk factor of depression and SI/SB in the context of long COVID-19.”

Reviewer 2

1.This manuscript is very brief for a review article, and really does not delve into sufficient detail of the articles included in the review. As a specific example, in section 3, the author only very briefly summarize the body of literature in support of astrocyte dysfunction as a driver of depression and suicidality. The section neglects important details including how the astrocytes were examined (i.e. by radiolabeled imaging, brain biopsy, postmortem brain examination, etc.) and what the comparison group was.

Thank you for this constructive feedback. We acknowledge the presence of the vast body of literature on astrocyte dysfunctions in depression and suicide, which has not been fully captured in our manuscript. However, details of such studies have been comprehensively reviewed elsewhere (Yamamoto et al., IJMS 2024; O'Leary et al., Glia 2021). The goal of our review is to present a concise distillation of such findings into mechanistic framework that enables further hypothesis-driven clinical exploration of the role of astrocytes in COVID-19-associated depression and SI/SB. We apologize for this brevity in the descriptive sections of astrocyte dysfunction in depression and suicide and have provided two tables with details of methodological approaches and control groups in all cited studies in these neuropsychiatric conditions (Tables 1-2). An introductory note on the primary goal of our review was also provided to clarify this issue.

“Clinical evidence for various astrocyte perturbations in suicide development is de-rived primarily from studies involving suicidal subjects and individuals with depression, a major risk factor for suicide. While details of such studies have been extensively re-viewed elsewhere [24, 25], this section aims to provide a concise distillation of relevant findings into mechanistic framework that enables further hypothesis-driven clinical exploration of the role of astrocytes in suicide development.”

2.A specific point of concern is that the authors incorrectly summarize the work of Bernstein et al (2015), stating that they found that "A specific astrocyte subset expressing glutamine synthetase (GLUS), an important enzyme involved in the synthesis pathway to prevent neuroexcitotoxicity, exhibited reduced density in various cortical brain regions of 151 depressed subjects, excluding gray matter areas [24]." Bernstein et al actually found that GLUS-expressing astrocytes were reduced in cortical gray matter (but not subcortical gray matter) of patients with major depression compared to patients with no mood disorder. This is not simply a semantic difference - the authors of this current manuscript, by stating 'excluding gray matter areas' suggest that GLUS-expressive astrocytes were reduced in the white matter only, which is not what Bernstein et al found. In addition, Bernstein et al were very specific about which brain regions were examined and separated these out by Brodmann Area. The authors of this current manuscript have inappropriately over-summarized this under the umbrella term 'various'. In general, the authors should take particular care to correctly state the findings of the articles included in the review and avoid over-summarizing.

Thank you for this constructive comment. We sincerely apologize for misquoting the work of Bernstein and colleagues. This mistake occurred in the language editing process of our manuscript, during which the word “subcortical” was erroneously removed. We have amended the text with details of brain regions examined in this study. We have also reviewed all cited works to ensure accuracy of the review and avoid over-summarizing/generalizing of nuanced findings.

“… several cortical (dorsolateral prefrontal, subgenual anterior cingulate, and anterior insular), but not subcortical, gray matter regions of depressed subjects compared to healthy controls [26] (Table 1).”

3.In section 4, the authors provide an overview of the body of literature on astrocyte dysfunction and COVID-19, which primarily focuses on GFAP. The authors have not constructed a compelling connection between the information presented in section 3 (astrocyte dysfunction is associated with depression/suicide) and the information presented in section 4.

Thank you for this helpful comment. We have revised the introductory remarks of the section on astrocyte dysfunction and COVID-19 to ensure a logical flow.

“Similar to depression and suicide, several astrocyte-related dysfunctions have been observed in the context of SAR-CoV-2 infection. As such, these abnormalities might represent an overlapping pathology in the development of these aforementioned neuropsychiatric conditions and COVID-19.”

  1. In addition, it is important to acknowledge that neuroinflammation is seen in a number of chronic conditions, and is certainly not specific to long COVID.

Thank you for this helpful comment. We have amended the text to note that the possible increase in suicide in long COVID-19 may result from prototypical neuroinflammation, which also occurs in other illnesses and is not specific to long COVID-19.

“…prototypical neuroinflammation, emanating from other cellular sources rather than reactive astrocytes, might represent the primary driver of depression and SI/SB. In fact, depression and suicide have been linked to other chronic neurobiological conditions [120-121], which share this pathological feature of neuroinflammation with long COVID-19”

Reviewer 3

1.Please, add Table with the discussed biomarkers in your review to compare them across acute vs long-term COVID patients

Thank you for this helpful suggestion. We have provided the summary of the discussed biomarkers in the Table 3 of the revised manuscript.

2.How specific COVID-19 infection to trigger suicide/depression in humans? Also, it is worth to add paragraph about specificity of COVID-19 vs other SARS-related infections to affect BBB-integrity and provoke suicide/depression. 

Thank you for this constructive comment. We have provided some description about this matter in the revised manuscript as follows:

“Of important mechanistic consideration, the potential increase in suicide among long COVID-19 patients might not be specific to SARS-CoV-2 infection. Firstly, several pathogens, ranging from Toxoplasma gondii and H1N1 to other coronaviruses (MERS-CoV and SARS-CoV-1), have been linked to increased depression and/or suicide in geographically distinct populations [112-116]. While a consensus among these association studies has not been reached [117-119], pathogen-induced BBB leakage might occur in different types of infectious illness with documented neurotropism, and thus, accounting for the possible increased in depression and suicide among the infected. Future imaging analysis of BBB integrity in patients experiencing neurological symptoms from these types of infection is warranted to confirm this hypothesis.”

  1. Do we have pre-clinical studies on COVID-induced depression? Please discuss if depressive patients are more prone to COVID-10 infection ?

Thank you for this constructive comment. We have revised the manuscript to address these issues as follows:

“Furthermore, some neuropsychiatric risk factors of SB, such as MDD, were recently linked to a subset of long COVID-19 patients [102, 103], suggesting that long COVID-19-associated depression might increase SB. In this regard, mid- to long-term studies (6-9 months) of long COVID-19 patients in France (n=60) and Brazil (n=425) revealed a prevalence of depression exceeding 30% in the French study [102] and 8% in the Brazilian study [103]. In addition, depressive symptoms reportedly persisted beyond 6 months in COVID-19 long haulers [104], with possible association to female sex [105]. These observations were further supported by a clinical observation covering six nations, which revealed an increased risk of depression at 16 months after COVID-19 infection among over 240,000 patients [106], and preclinical findings of depressive behaviors in mice triggered by COVID-19/SARS-CoV-2 infections [107, 108]. However, it’s worth noting that depressed subjects reportedly did not exhibit increased susceptibility to COVID-19 infection [109, 110].”

  1. It would be interesting to find out if certain genetic predisposition to depression/suicide, affects the immune system and, as result COVID-19 infection could exacerbate depressive symptoms in such population of humans. 

Thank you for this constructive comment. We have provided some comments on this very interesting topic as follows:         

“.. no genetic predispositions, including those affecting inflammatory signaling and the immune system, have been linked to depression and suicide in the context of COVID-19 [122].”

  1. As depression is accompanied by changes in brain' neurotransmitters, e.g. 5-HT, authors should also discuss effects of COVID-19 on brain' neurotransmitters to discuss it in terms of the classical hypothesis of the depression.

Thank you for this constructive comment. We have amended the text to describe the classical hypothesis of depression in the context of long COVID-19 as follows:

“Lastly, increased depression and suicide in long COVID-19 might result from dysregulation of pathways other than those of inflammatory origins. In support of this alternative view, no genetic predispositions, including those affecting inflammatory signaling and the immune system, have been linked to depression and suicide in the context of COVID-19 [122]. Furthermore, lower level of serotonin has recently been linked to long COVID-19 [123], prompting the possibility that defective serotonergic transmission might represent an underlying mechanism of depression and SI/SB [124] in the context of this infectious illness.”   

Reviewer 2 Report

Comments and Suggestions for Authors

The authors present a review of the sizable body of literature on the association between astrocyte dysfunction and depression/suicide and also present a review of the literature on the association between astrocyte dysfunction and COVID. They conclude by suggesting that astrocyte dysfunction may be a mediator in the relationship between COVID and depression/suicide.

This manuscript is very brief for a review article, and really does not delve into sufficient detail of the articles included in the review. As a specific example, in section 3, the author only very briefly summarize the body of literature in support of astrocyte dysfunction as a driver of depression and suicidality. The section neglects important details including how the astrocytes were examined (i.e. by radiolabeled imaging, brain biopsy, postmortem brain examination, etc.) and what the comparison group was.

A specific point of concern is that the authors incorrectly summarize the work of Bernstein et al (2015), stating that they found that "A specific astrocyte subset expressing glutamine synthetase (GLUS), an important enzyme involved in the synthesis pathway to prevent neuroexcitotoxicity, exhibited reduced density in various cortical brain regions of 151 depressed subjects, excluding gray matter areas [24]." Bernstein et al actually found that GLUS-expressing astrocytes were reduced in cortical gray matter (but not subcortical gray matter) of patients with major depression compared to patients with no mood disorder. This is not simply a semantic difference - the authors of this current manuscript, by stating 'excluding gray matter areas' suggest that GLUS-expressive astrocytes were reduced in the white matter only, which is not what Bernstein et al found. In addition, Bernstein et al were very specific about which brain regions were examined and separated these out by Brodmann Area. The authors of this current manuscript have inappropriately over-summarized this under the umbrella term 'various'. In general, the authors should take particular care to correctly state the findings of the articles included in the review and avoid over-summarizing.

In section 4, the authors provide an overview of the body of literature on astrocyte dysfunction and COVID-19, which primarily focuses on GFAP. The authors have not constructed a compelling connection between the information presented in section 3 (astrocyte dysfunction is associated with depression/suicide) and the information presented in section 4. In addition, it is important to acknowledge that neuroinflammation is seen in a number of chronic conditions, and is certainly not specific to long COVID.

Author Response

(The authors gave the same response as above.)

Reviewer 3 Report

Comments and Suggestions for Authors

Dear Alessandra Costanza and co-authors,

Your review "Reactive astrocytosis as contributor to increased depression and suicide risk in long COVID-19?" is timely and raises an important question about mental health in post-covid patients. I have several suggestions which could improve your review, including:

1. Please, add Table with the discussed biomarkers in your review to compare them across acute vs long-term COVID patients;

2. How specific COVID-19 infection to trigger suicide/depression in humans? Do we have pre-clinical studies on COVID-induced depression? Please, discuss it. Also, it is worth to add paragraph about specificity of COVID-19 vs other SARS-related infections to affect BBB-integrity and provoke suicide/depression. 

3. Please, add paragraph to discuss if depressive patients are more prone to COVID-10 infection (?). It would be interesting to find out if certain genetic predisposition to depression/suicide, affects the immune system and, as result COVID-19 infection could exacerbate depressive symptoms in such population of humans. 

 4. As depression is accompanied by changes in brain' neurotransmitters, e.g. 5-HT, authors should also discuss effects of COVID-19 on brain' neurotransmitters to discuss it in terms of the classical hypothesis of the depression.

Author Response

(The authors gave the same response as above.)

Round 2

Reviewer 2 Report

Comments and Suggestions for Authors

The authors have addressed the reviewer concerns.

Comments on the Quality of English Language

Minor language editing required